# First Evidence Indicates the Physiology- and Axial-Myopia-Dependent Profiles of Steroid Hormones in Aqueous Humor

**DOI:** 10.3390/metabo12121220

**Published:** 2022-12-05

**Authors:** Tiansheng Chou, Xiaosheng Huang, Jiawei Liu, Xinhua Liu, Kun Zeng, Zonghui Yan, Shaoyi Mei, Liangnan Sun, Wenqun Xi, Jinglan Ni, Jin Zi, Jun Zhao, Siqi Liu

**Affiliations:** 1College of Life Sciences, University of Chinese Academy of Sciences, Beijing 100049, China; 2Department of Proteomics, Beijing Genomics Institute (BGI), Shenzhen 518000, China; 3Shenzhen Eye Institute, Shenzhen Eye Hospital, Jinan University, Shenzhen 518040, China; 4Department of Ophthalmology, Shenzhen People’s Hospital (The Second Clinical Medical College, Jinan University, The First Affiliated Hospital, Southern University of Science and Technology), Shenzhen 518020, China

**Keywords:** steroid hormone, aqueous humor, blood–aqueous barrier (BAB), axial length (AL)

## Abstract

The quantitative level of steroid hormones (SHs) in some body fluids have been accepted for clinical diagnosis, whereas their distribution in aqueous humor (AH) is unknown yet. Herein, a profiling study was conducted with a total of 171 AH and 107 plasma samples using liquid chromatography coupled with tandem mass spectrometry (LC MS/MS). For the first time, six kinds of SHs in AH were quantitatively estimated, and their abundances were ranked at cortisol (F), corticosterone (COR), androstenedione (A2), and 11-deoxycortisol (11DOC). The corresponding abundance of all SHs in AH was significantly lower than those in plasma, while there was a lack of a proportional relationship with the abundance of plasma SHs. Dehydroepiandrosterone sulfate, the most abundant plasma SH, was undetectable in AH, implying that the blood–aqueous barrier might specifically block its transferral. Axial myopia generally results from many factors throughout the entire eye from tissues and molecules; furthermore, the correlation of AH SHs and axial myopia was assessed to look for their indication in such myopia. The panel with five kinds of AH SHs (F, COR, CORT, ALD and A2) was functional as a discriminator for axial myopia and control. The abundance of SHs, therefore, has a specific distribution in AH and can potentially contribute to axial myopia.

## 1. Introduction

Steroid hormones (SHs) are a group of compounds that share a cyclopentan-o-perhydrophenanthrene ring derived from cholesterol [1]. Depending on the secretory organs, these compounds are categorized into two major classes (corticosteroids and sex hormones) even though some kinds of SHs are generated by the placenta during pregnancy [2]. By regulating a series of physiological activities, it is well-known that SHs are tightly associated with diseases [3,4,5,6,7]. For example, the plasma of patients with Conn’s syndrome contains high concentrations of aldosterone (ALD), and that of patients with Cushing’s syndrome holds high concentrations of cortisol (F) [8,9]. As the pathways of SH synthesis and metabolism are complicated, a single SH may not truly reflect the relationship between hormone disorders and diseases in most cases. Deborah and Stefan [10] claimed that simultaneous multiple hormonal imbalances of glucocorticoids, mineralocorticoids or sex hormones were found in the plasma of patients with congenital adrenal hyperplasia (CAH), while Ye et al. [11] constructed a discriminator for 3 CAH subtypes using 13 SHs in blood. Biomarkers consisting of multiple SHs were not only employed for biochemical analysis in plasma or serum samples but also were discovered in other body fluid and tissue. Vasileios et al. [12] evaluated the risk of postoperative recurrence in adrenocortical carcinoma based on the abundance of 19 kinds of SHs in urine using random forest prediction, while Sosvorova et al. [13] found that normal-pressure hydrocephalus could be predicted by combinations of SHs in cerebrospinal fluid. The profiling of SHs in body fluids has emerged as an important indicator related to the status of health and disease.

The SH abundance in blood was partially associated with some eye diseases. Rosa et al. [14] reported higher concentrations of SHs in the serum samples of patients with chronic central serous chorioretinopathy, such as ALD, estrone (E1), etiocholanolone and androstenedione (A2). Tina et al. [15] observed that sex hormone changes could affect tear film composition and function along with ocular surface structures and components. Aqueous humor (AH) is an intraocular fluid containing metabolites and proteins that fills the anterior and posterior chamber, provides nutrients to avascular tissues and removes metabolic waste from the intraocular space [16]. Some eye diseases are related to the SH levels in blood [14,15]. The mechanism by which SHs in AH, as an intraocular microenvironment, contribute to eye diseases is still unknown. Zhang et al. [17] measured estrogen and progesterone (P) in human AH using enzyme-linked immunosorbent assay and found that the hormone levels in AH were lower than those in serum with a gender-independent mode and did not find a difference between cataracts and non-cataracts. Many studies revealed that high intraocular pressure could be attributed to locally or systemically administered glucocorticoids [18,19,20,21,22]. Porter and Silber [22] first claimed that topical steroid therapy in AH could increase the risk of higher intraocular pressure by more than long-term systemic therapy in serum. Several studies documented endogenous SHs in AH potentially related to eye diseases, while there has been a lack of sufficient evidence to indicate the profile of SHs in AH and the abundance rank of these hormones. With the limitation of knowledge of the SHs of AH, it is difficult to deepen our understanding of the physiological roles of SHs in AH and eye diseases and to develop a proper ophthalmologic therapy related to SHs.

This study established an LC MS/MS-based approach to globally profile and quantify the SHs in AH and addressed the following issues: (1) the distribution of endogenous SHs in AH, (2) the relationship between SHs in blood and AH, and (3), as axial myopia is believe the results of microenvironment changes in entire eye and is assumed to be due to some interference from metabolites, the correlation of AH SHs and axial myopia was evaluated to understand the potential role of SHs in AH as biomarkers. Over one hundred AH and plasma samples were collected from patients with cataracts or axial myopia, and their content of SHs were examined. The statistical discriminator was employed to predict the clinical values of SHs in AH.

## 2. Materials and Methods

### 2.1. Chemicals

The standard SHs were obtained mainly from commercial sources: 11-deoxycorticosterone (DOC), A2, corticosterone (CORT), F, dihydrotestosterone (DHT), P and testosterone (T) from Dr. Ehrenstorfer GmbH (Augsburg, Germany); 17-hydroxyprogesterone (17OHP), cortisone (COR), E1 and pregnenolone (Pr) from Sigma Aldrich (St. Louis, MO, USA), 11-deoxycortisol (11DOC), 17-hydroxypregnenolone (17OHPr), ALD and estradiol (E2) from Toronto Research Chemicals (Toronto, ON, Canada); 21-deoxycortisol (21DOC), dehydroepiandrosterone (DHEA) and dehydroepiandrosterone sulfate (DHEAS) from Cerilliant (Round Rock, TX, USA); and estriol (E3) from Aladdin (Shanghai, China). The standard SHs labeled with isotopes were also obtained from commercial sources: 11-DOC-d5, 17OHP-^13^c3, Ald-d8, A2-^13^c3, DHEA-d6 and T-^13^c3 from Cerilliant (Round Rock, TX, USA); 17OHPr-^13^c2-d2, 21DOC-d8, DHEAS-d6, E3-^13^c3, Pr-d4 and P-^13^c3 from Sigma Aldrich; DOC-d8, E2-d3 and E1-d4 from Toronto Research Chemicals (Toronto, ON, Canada); and F-d4 and COR-d7 from CDN Isotopes (Pointe-Claire, QC, Canada). Double-charcoal-stripped human serum was purchased from BBI Solutions. LC–MS-grade methanol (MeOH) and acetonitrile (ACN) were obtained from Fisher Scientific (Waltham, MA, USA). Methyl tert-butyl ether (MTBE) and ammonium fluoride were purchased from Aladdin (Shanghai, China).

### 2.2. Clinical Samples

The participants were recruited from the patients who underwent cataract surgery or myopia surgery at Shenzhen Eye Hospital (Shenzhen, China) from July 2020 to December 2021. A total of 171 AHs from individual patients were divided into four types based on clinical diagnoses: 47 with age-related cataracts (ARC), 45 with high-myopia cataracts (HMC), 52 with high myopia (HM) and 27 with low myopia (LM). All study subjects received axial-length evaluation with a ZEISS IOL Master 700. Based on the ophthalmologic criteria [23,24], the donors were divided into two groups: a myopia group (axial length (AL) ≥ 24 mm) and a control group (AL < 24 mm), in which 132 were myopic (78 females and 54 males) and 39 were control (24 females and 15 males). None of the donors had received steroid hormone treatment in the past 6 months, and the biochemical parameters of blood remained in the normal range. All of the myopia donors were diagnosed as axial myopia but not refractive and secondary myopia. All of the donors were generally grouped in (1) the myopia group, 18–40 years old; (2) the high myopia cataract group, 18–60 years old; and (3) the age-related cataract group, over 50 years old. All of the donors were evaluated by clinical and ophthalmological diagnosis. The exclusion criteria were (1) other eye diseases such as glaucoma, uveitis and trauma; (2) metabolic diseases such as hyperlipidemia and hyperuricemia; or (3) severe diseases, such as rheumatic immune diseases, tumors and genetic diseases. The clinical information of these patients is recorded in Table 1. AH samples were collected from these participants during cataract or implantable collamer lens implantation surgery. Meanwhile, a sample of antecubital vein blood was also obtained from the same patients (Table 1). After operation, all samples were frozen immediately and stored at −80 °C until the next experiments. In accordance with the Declaration of Helsinki, this study was registered in the Chinese Clinical Trial Register (Trial number: ChiCTR2100042651) and was approved by the Ethics Committee of Shenzhen Eye Hospital (NO.: 20200618-11). Shenzhen Eye Hospital strictly followed the approved ethical procedure to collect all of the AH samples, and the donors were informed about the sample collection, and they have signed the informed consent forms.

### 2.3. Extraction of SHs from AH and Plasma

The protocol to extract SHs from AH was similar to that from plasma. A body fluid sample of approximately 100 μL was diluted with phosphate buffered saline (PBS) and mixed with 1500 μL ACN/MTBE (*v*:*v* = 1:9), followed by thorough vertexing and centrifugation at 4000× *g*. The resulting supernatant was carefully transferred to a glass tube and was dried in a nitrogen bath at room temperature. The dried extract was stored at −20 °C and reconstituted with 60% MeOH for SH analysis.

### 2.4. Identification and Quantification of SHs Using LC MS/MS

The reconstitution of SH extract in MeOH was uploaded onto an ACQUITY UPLC I-Class UPLC system (Waters, Wilmslow, UK) mounted with an ACQCLTY UPLC CSH C18 column (Waters, 2.1 × 100 mm, 1.7 µm). Mobile phase A was an aqueous solution containing 0.1 mM ammonium fluoride, and mobile phase B was MeOH. The elution gradient was held for 1.5 min and set as follows: within the period of 0–0.5 min, mobile phase B was from 20% to 40%; from 0.5–3.3 min, from 40% to 55%; from 3.3–6 min, from 55% to 95%. The elution was directly injected into a SCIEX Triple Quad™ 6500+ mass spectrometer (SCIEX, Framingham, MA, USA) equipped with an Ion Drive Turbo V source and electrospray source interface for the acquisition of MS/MS signals. The parameters of this machine were optimized: ion spray voltage at −4.5 kV/+5.5 kV (negative/positive), ion source gas 1 at 40 psi, ion source gas 2 at 40 psi, curtain gas at 35 psi and source temperature at 550 °C. The MS/MS signals of interest were alternatively monitored in positive and negative ion modes with a multiple reaction monitoring (MRM) approach. The mass transitions for the SHs in this study are listed in Appendix A. Data acquisition was performed using Analyst software (SCIEX, version 1.7), and Multi Quant software (SCIEX, version 3.0.2) was used for the detection and quantification of MS/MS peaks.

### 2.5. Quantification of SHs in AH and Plasma

All standards of SHs with/without isotopes were dissolved in MeOH as recommended by the manufacturer and were stored at −20 °C before analysis. To calibrate all SHs of interest, a stock solution containing 19 SHs without isotope labeling was prepared at final concentrations of ng/mL, 11DOC at 10, 17OHP at 25, 17OHPr at 80, 21DOC at 50, A2 at 10, ALD at 5, COR at 100, CORT at 30, DHEA at 30, DHEAS at 4000, DHT at 20, DOC at 5, E1 at 8, E2 at 20, E3 at 10, F at 250, P at 20, Pr at 80 and T at 20. To prepare the working solution of the internal standard (IS), a mixture of the standards of SHs with isotope labeling was made at final concentrations of 17OHP, P and T at 1 ng/mL; 11DOC, 21DOC, A2, COR, CORT, DHEAS, DHT, DOC, E1, E2 and E3 at 10 ng/mL; 17OHPr, ALD, DHEA and F at 100 ng/mL; and Pr at 500 ng/mL. The IS working solution was accompanied by all measures including calibration, quality control and sample analysis. For the quantification of the AH SHs, the MRM signals responding to 8 different concentrations of the hormones in methanol and a fixed concentration of the isotope labeled hormones were recorded, and the ratios of MRM signals from the varied concentrations of hormones against that from the fixed concentration of isotope-labeled hormones were calibrated versus the hormone concentrations. For the quantification of plasma SHs, the calibration was made on a matrix background from double-charcoal-stripped human serum. To quantify the SHs in AH or plasma, MRM signals related to the endogenous and spiked target SH were acquired, and the ratios of the two MRM signals were generated, which were used to estimate the SH quantities based on calibration curves made by standard and isotope-labeled SHs.

### 2.6. Quality Control of MRM Signals to Monitor SHs

To evaluate whether AH could exert a matrix effect on SH quantification using LC MS/MS, an equal IS working solution was added to AH and MeOH, while the MRM responses to the two cases were statistically assessed for significant differences. According to the Guidance for Industry on Bioanalytical Methods Validation (Food and Drug Administration, Silver Spring, MD, USA) [25] principle, the matrix effect is regarded as negligible or considerable. A recovery test was conducted to compare the changes in MRM signals responsive to standard hormones that were treated with/without the extraction procedure. Imprecision was estimated as the bias of the known concentration and measured concentration with 8 concentrations of a hormone in triplicate.

The limit of quantification (LOQ) was determined as the lowest concentration based on the calibration curve for a hormone with ±20% tolerance. The limit of detection (LOD) was defined as the lowest concentration of a hormone that could be consistently detected. Reproducibility was evaluated through the measurements in inter- and intra-assays of the MS signal output with the coefficient of variation to judge the reproducibility in assays.

### 2.7. Proteomic Analysis of AH and Plasma

Samples of AH and plasma from a cohort containing 25 individuals (16 cataract and 9 non-cataract) were collected, and their proteins were extracted individually. The extracted proteins were individually digested by trypsin, and the digested peptides were delivered to an LC MS/MS Fusion Lumos tandem mass spectrometer (Thermo Fisher Scientific, San Jose, CA, USA). The MS/MS signals corresponding to the peptides were acquired in data-independent acquisition mode (DIA). The DIA data were treated with Spectronaut (version X) for protein identification and quantification. The details of such proteomics analysis are described in another manuscript (in preparation), while only the proteomics data related to corticosteroid-binding globulin (CBG) and sex-hormone-binding globulin (SHBG) were introduced in this study.

### 2.8. Statistical Analysis for AH SHs

The R program language (version 4.1.2) was used for all data analyses. Following Shapiro–Wilk’s test for normality, unpaired Wilcoxon nonparametric tests were used to determine the significance of abundance differences between plasma and AH (ggpubr (version 0.4.0)). The abundance correlations among the individual SHs in AH were estimated by Spearman’s correlation analysis with the R package ggcorr (version 1.5.0). The discrimination of AL based on the steroid hormone abundance in AH was performed using the support vector machine (SVM) model with R package e1071 (version 1.7-11). All plots for the statistical analysis were drawn with the R package ggplot2 (version 3.3.6). The results of SHs distribution of concentration in cohort were expressed as mean ± SD. In general, 25 subjects were basically required for a power analysis (effect size: 0.8). A total of 171 AHs from individual patients were divided into four types based on clinical diagnoses: 47 with age-related cataracts (ARC), 45 with high-myopia cataracts (HMC), 52 with high myopia (HM) and 27 with low myopia (LM).

## 3. Results

### 3.1. Assessments toward the Quantification of SHs in AH

To accurately quantify AH SHs, the quality control in the LC MS/MS experiment was conducted in five aspects: matrix effect, concentration calibration, limit of quantification, extraction recovery and experimental intra- or inter-precision.

Whether AH influences the MS signals derived from SHs was assessed by monitoring the changes in MS signals corresponding to 1–500 ng/mL of 19 kinds of SHs labeled by isotopes between AH and 60% MeOH aqueous solutions. The matrix effects of AH on the different SHs are presented in Figure 1a. The top five matrix effects of AH were observed in DOC, T and A2 (approximately 80%) and Pr and P (approximately 120%), while the low matrix effects were perceived in 14 kinds of hormones, such as E3, DHT, DHEAS, 17OHPr, 11DOC, CORT, COR, E2, 21DOC, E1, F, DHEA, 17OHP and ALD, approximately 100%. As the MS signals of most SHs were lightly affected by AH, the AH matrix effects on a few other SHs were restricted within a limit range of 80 to 120%. The matrix effect caused by AH was assumed to be negligible; thus, quality control experiments were performed in a solution of SHs dissolved in aqueous methanol [25].

The quantitative calibrations were generated from the MS intensities responding to eight series dilutions of 19 kinds of SHs. The calibration results are illustrated in Appendix A, indicating that all hormones retained good linearity at the ng/ml level with a correlation coefficient (R^2^) over 0.99 and a coefficient of variation less than 15% (except for 11DOC). Following the calibration curves, the limit of quantification (LOQ) for these hormones were determined (Appendix A). Note that the quantitative detections for the six kinds of SHs that are mentioned in the next sections (F, COR, CORT, A2, ALD and 11DOC) were limited from 3 to 100 pg/mL. Based on the calibrations for these hormones, the second highest (high) and lowest (low) concentrations of each individual hormone were selected to test the extraction recovery. As depicted in Figure 1b, the extraction recoveries for all 19 kinds of hormones were at the acceptable level of approximately 80–120%, while the recoveries at higher concentrations were basically comparable with those at lower concentrations. Specifically, the average recoveries for the six kinds of hormones described later were 93.8 ± 3.8% (low) and 94.1 ± 4.2% (high), respectively.

With a similar strategy to select the concentrations of SHs above, the two concentrations from each individual hormone were taken for the evaluation of quantitative precision and reproducibility in intra- or inter-assays. Appendix A summarizes the evaluation results. The intra-assay accuracies ranged from −8 to 7% (high) and −5 to 11% (low), while the inter-assay accuracies ranged from −8 to 5% (high) and −3 to 14% (low). The intra-assay coefficient of variation ranged from 1 to 10% (high) and 2 to 13% (low), while the inter-assay coefficient of variation ranged from 2 to 20% (high) and 3 to 14% (low). The precision and reproducibility for these hormones at high concentrations, either in intra- or inter-assay, were generally better than those at low concentrations, except for Pr. The values of quality control for LC MS/MS data were well accepted in the quantification of SHs. Moreover, the six kinds of SHs mentioned latter exhibited better performance in terms of both precision and reproducibility, with accuracies ranging from −4 to 4% (high) and −5 to 7% (low) in the inter-assay and 2 to 7% (high) and −4 to 6% (low) in the intra-assay, while the coefficient of variation spanned 6 ± 3% (high) and 7 ± 3% (low) in the inter-assay and 5.7 ± 3% (high) and 7 ± 2% (low) in the intra-assay. Higher values of quality control in experiments from the extraction to the calibration of SHs in AH, therefore, were satisfied by accurate quantification.

### 3.2. Profiles of SHs in AH

Following the LC MS/MS method and the parameters of quality control mentioned above, the levels of endogenous SHs from total of 171 AH samples collected were qualitatively and quantitatively measured. Several SHs listed in Appendix A were detected, such as F, COR, CORT, A2, ALD, 11DOC, T, E2, E1 and DHT. However, some SHs were undetectable in AHs, such as DHEAS, DHEA, 17OHPr, Pr, P, 17OHP, DOC, 21DOC and E3. A criterion was set in this study to define the AH SH: its concentration in AH ≥ LOQ and detection frequency in the cohort ≥ 50% [26]. The profile of SH in AH, therefore, was generated, including F, COR, CORT, A2, ALD and 11DOC. The six AH SHs in the cohort have no missing values. The quantitative rank of this panel is illustrated in Figure 2a, F placed at the top and A2, ALD and 11DOC at lower and comparable levels. Detailed information on the quantitative measurement of endogenous SHs in AH is listed in Appendix A, showing that the concentrations of AH SHs ranged over three orders of magnitude.

The relevance of the AH SHs and physiologic parameters was assessed in two aspects: gender and age. As shown in Figure 2b and Appendix A, the AH SHs were generally divided into two groups: gender-independent, such as COR, CORT, ALD and 11DOC, and gender dependent, such as F and A2. A2 is a sexual hormone generally with higher concentrations in males than in females. The fact that the A2 concentrations in male AHs were higher than those in female AHs appeared understandable; nevertheless, the higher concentrations of F in male AHs raised a question. In this cohort, F remained the top concentration compared with the other five hormones, while in biochemical pathways, F seems relatively independent from the generation of sexual hormones (Figure 2d). Hence, the observation of higher F concentrations in male AHs might lead to a research direction to explore the gender dependency of F in AH, biological significance and functions. Regarding whether the AH SHs are age dependent, the AH samples in this cohort were broadly divided into three groups according to age classification: <30-year-olds, constituting 51 donors; 30–60-year-olds, constituting 67 donors; and ≥60-year-olds, constituting 53 donors. The average concentration differences of each hormone among these groups were statistically evaluated using the average values of the young group (age <30 years old) as references (Figure 2c and Appendix A). The concentrations of all AH SHs between the <30-year-old and 30–60-year-old groups were comparable, whereas the concentrations of the three hormones COR, A2 and ALD in the ≥60-year-old group were substantially lower than those in the other two groups (Figure 2c). It is generally accepted that the sex hormone abundance in serum decreases with aging [27,28]. If the A2 in AH is gained from its diffusion from serum, then the lower concentrations of A2 in the AH group with age ≥60 years seem understandable. Lower concentrations of COR and ALD in aged AH, however, were first observed in this study. The relevant consequence of this event in physiology merits great attention.

The abundance correlation among the AH SHs was further appraised through their synthetic pathways. As presented in Figure 2d, the corresponding biosynthesis of SHs mainly from the adrenal cortex and sex gland originates from cholesterol and then travels in two pathways: the generation of corticosteroids and sexual hormones. In AH, over 80% of the quantitated SHs (5/6) were located in corticosteroid biosynthesis, while only one sex hormone, A2, was from another synthesis pathway of SHs. Moreover, three AH SHs with higher abundance, F, COR and CORT (Figure 2a), are located in corticosteroid biosynthesis. The abundances of F, COR and ALD were moderately correlated with each other according to Spearman’s correlation coefficient (R > 0.75), while the abundances of CORT, A2 and 11DOC were poorly correlated with each other (Figure 2e). The AH hormones with higher abundance correlations were the final products of corticosteroid biosynthesis, and 11DOC or CORT, which is the up metabolite of F or ALD, was measurable but at low abundance. Moreover, other early intermediates in corticosteroid biosynthesis, such as Pr, P, DOC, 17OHPr and 17OHP, were undetectable or in very low abundance in AH. The biosynthetic components of corticosteroid pathway in AH seem to be incomplete, leading to a postulation that these AH hormones may be gained from the metabolite transportation of serum but are not produced through AH-related eye tissues. As regards the biosynthetic pathway of sexual hormone, A2 exhibited similar situation because its up metabolite, DHEA, was not found in AH using LC MS/MS.

### 3.3. Comparison of SH Profiles between AH and Plasma

It is well-known that the bloodstream acts as a carrier of SHs and transports them to different target sites. The SH transfer method from serum to other body fluids is partially understood, such as cerebrospinal fluid crossing the blood–brain barrier. Nevertheless, the mechanism of SHs transported from plasma to AH through the blood–aqueous barrier is still not understood. If the AH SHs are indeed passively diffused from serum, the question is naturally raised as to whether the composition and abundance proportion of SHs in serum is similar to that in AH. Therefore, parallel measurements of SHs, both AH and plasma in individuals were conducted in this study. A total of 107 individuals donated both AH and plasma samples, and the quantitative measurements on average for SHs in AH (6 SHs) and plasma (17 SHs) are summarized in Appendix A. Figure 3a shows the profile of plasma SHs, in which 17 SHs were identified and quantified across five orders of magnitude, DHEAS at the top with 357 ng/mL and DOC at the bottom with 0.036 ng/mL. Obviously, there were more types of plasma SHs than AH SHs, whereas no specific SH was found in AH. A quantitative comparison of the co-identified hormones AH and plasma in this cohort is presented in Figure 3b. In view of the comprehensive assessment of the information on AH and plasma SHs, there were three key results: (1) DHEAS with the highest abundance and DHEA with relatively high abundance in plasma measured by current LC MS/MS technology were undetectable in AH, suggesting something in the blood–aqueous barrier blocking DHEAS and DHEA transferred from plasma to AH; (2) the abundance order of the AH SHs was similar to their order in plasma, supporting the hypothesis of the AH hormones being sourced from plasma; (3) if the transfer of the plasma SHs to AH was in a simple diffusion mode, the ratios of hormone abundance between plasma and AH should be relatively consistent. However, such ratios were flexible (plasma/AH in Figure 3b), implicating that the transfer process of such hormones between AH and plasma is likely to be specifically selective.

Evaluation of the equilibrium status of SHs between AH and plasma could be implemented in various aspects, including an individual comparison of the hormone abundance between AH and plasma. If the transfer of SHs through the blood–aqueous barrier is based on simple diffusion, the higher abundance of SHs is found in an individual sample of plasma, and the more hormone is detected in the corresponding AH. To test this assumption, the abundance of the six AH SHs in an individual sample of plasma was ranked first, then the corresponding SH abundance in the same sample was checked whether it matched to the rank order. In Figure 3c, the abundance ranks of plasma SHs from all of the individual samples are depicted on the right sides (red), while the abundance of AH SHs simply matched to the individual samples are presented on the left sides (blue). The ranking orders of all the plasma SH abundance did not coincide with those of the abundance orders of AH hormones, reflecting that the abundance of SHs between plasma and AH was generally in disequilibrium but personally dependent at the individual level. The observation elicited from Figure 3b,c raises another question, on which kind of regulation can balance the abundance distribution of SHs between plasma and AH.

CBG and SHBG are well-accepted as hormone transporters in plasma [29]. Compared with albumin with nonspecific binding to hormones, CBG binds to natural glucocorticoids with high affinity, and SHBG, as a glycoprotein, specifically interacts with androgens and estrogens [30,31]. A question was raised as to whether the two SH binding proteins played a key role in hormone transport from plasma to AH. A total of 25 samples were taken for profiling and quantifying the proteomes and SHs of plasma and AH in parallel. As presented in Figure 3d, the level of CBG or SHBG in plasma was basically comparable to that in AH, whereas the abundance of CBG or SHBG in plasma was slightly higher than that in AH, 1.34-fold for CBG and 1.27-fold for SHBG on average. The cumulative abundance of corticosteroids and sex hormones in plasma was much higher than that in AH (plasma/AH for corticosteroid = 30.37-fold and plasma/AH for sex hormones = 8.58-fold, Figure 3e). If the transport of SHs was mainly regulated by CBG and SHBG, then the abundance of corticosteroid or sex hormones would be basically comparable in plasma and AH because of the relatively equal abundance of CBG and SHBG in the two body fluids. In fact, the two hormone types in AH were found to have significantly lower abundance than those in plasma. Moreover, the correlation assessment for the abundance of corticosteroid or sex hormone and their corresponding CBG or SHBG in individual plasma and AHs revealed a poor correlation between the abundance of SHs and their carrier proteins (Appendix A). The inappropriate abundance correlation of SHs and their carrier proteins in plasma and AH prompts a clue that the regulation of SH transfer from plasma to AH does not uniquely rely on CBG and SHBG but is partially controlled by the blood–aqueous barrier, further study of which is urgently required in future studies.

### 3.4. Abundance Responses of AH SHs to Axial Myopia

Myopia is a common eye disease, which is generally divided into three subtypes, axial, refractive and secondary myopia. Compared with refractive and secondary myopia that is caused by the pathological changes in local tissues such as the cornea or lens and the consequence of another disease, axial myopia is generally believed as the result of the distorted distance from the corneal surface to the retinal pigment epithelium in the entire eye, which is impacted by many factors such as degeneration, oxidative stress and metabolites. As the AH components can exert a lot of influence on whole-eye tissues, axial myopia is used to evaluate the AH effect on axial abnormality. In this study, we only focused on the correlation of the SH quantitative profile and axial myopia to look for the potential indicator(s) of such myopia at the molecular level.

The relationship of the AH SH levels and axial myopia was carefully examined in the cohort with 171 AH samples, including 92 individuals with cataract and 81 individuals with a clear lens. The clinical information for these donors is summarized in Table 1. Based on the ophthalmologic criteria, all study subjects were divided into two groups: a myopia group (axial length (AL) ≥ 24 mm) and a control group (AL < 24 mm), in which 132 were myopic (78 females and 54 males) and 39 were control (24 females and 15 males). None of the participants had received steroid hormone treatment in the past 6 months, and the biochemical parameters of blood remained in the normal range. As shown in Figure 4a, the abundance of four kinds of SHs (COR, CORT, ALD and A2) was significantly higher in the myopia group than in the control group, whereas two kinds of SHs (F and 11DOC) were not different between the myopia and control groups. This is the first observation to imply the potential correlation of SHs in AH with myopia. Importantly, statistical evaluation also suggested no significant correlation of AH SHs between patients with cataracts and non-cataract patients (Appendix A). The significance with a Student’s *t* test could be used to judge an acceptable change in the abundance of SHs among the clinical samples; however, the judgment was not useful for clinical application. Therefore, the SVM algorithm was employed to determine a model that enabled the discrimination of myopia from normal based on the abundance of AH SHs. In this cohort, the different levels of SH abundance were iteratively examined by SVM to look for the capacity to discriminate between axial myopia and normal vision and between single and combined SHs, and the best values of receiver operating characteristic at each level were determined by area under the curve (AUC) and F1 score that are the harmonic means of the precision and sensitivity (Figure 4b). In all discriminative predictions by a single AH SH, A2 achieved the best performance, with an AUC of 0.809 and an F1 score of 0.653, and COR was second-ranked, with an AUC of 0.756 and an F1 score of 0.670. The combination of A2 and COR improved the prediction, with an AUC of 0.826 and an F1 score of 0.731. Adding more hormones to the discriminators based on A2 and COR gained better prediction until the combination of hormones reached five. The panel consisting of five AH SHs, A2, COR, F, CORT and 11DOC, was therefore concluded by SVM analysis to be the best discriminator for myopia, with an AUC of 0.911 and an F1 score of 0.797. This is the first evidence that reveals a profile of AH SHs as an indicator of eye abnormality.

## 4. Discussion

SHs are mainly synthesized in adrenal glands and gonads and are delivered to their target tissues/organs through blood circulation. Hence, it is inferred that the SHs in AH come from blood. A comparison study of the SH distribution in plasma and SHs is shown in Figure 3, demonstrating that the types and quantities of SHs in AH were very different from those in plasma. First, DHEA and DHEAS, which are SHs with high abundance in human plasma, were almost undetectable in human AHs. DHEA and DHEAS are important SHs mainly of adrenal origin and originate partially from the gonads and brain. The two SHs in plasma can easily cross the brain–blood barrier through the special transporter system. Asaba et al. [32] discovered that organic anion transporting polypeptide 2 (OATP2), located in the brain–blood barrier, could specifically mediate the efflux of DHEA and DHEAS, while DHEA and DHEAS enter the brain and enable potent modulators of neural function, including neurogenesis, neuronal growth and differentiation and neuroprotection. The absence of DHEA and DHEAS in human AH implies a lack of the appropriate transporter on the blood–aqueous barrier. Second, the quantitative distribution of the six SHs between AH and plasma did not follow a certain proportional mode because the abundance ratio of a SH in AH to plasma in general (Figure 3b) or in individuals (Figure 3c) was inconsistent, suggesting that the transfer process of SHs from plasma to AH was mediated by some regulators rather than passive diffusion. Third, the abundance ratios of SH carrying proteins CBG and SHBG were not proportional to the abundance ratios of corticosteroids and sex hormones (Figure 3d), indicating that the SH carry proteins did not play a sole role in SH transfer from plasma to AH. Taking all of the evidence together, the question of whether a specific transfer system of SHs on the blood–aqueous barrier has been explored yet.

AL is the most significant index to represent the degree of axial refractive error [23,33]. Generally, axial myopia results from an increase in AL, and AL > 24 mm was the index for axial myopia in the adult myopic population [33]. Several myopia-related proteins in AH have been studied. For instance, matrix metalloproteinases (MMPs), metalloproteinases (TIMPs), transforming growth factor-β (TGF-β) and TGF-β2 in AH were positively associated with AL [34]. For metabolites in AH, Ji et al. [35] used an AH sample set containing 20 donors with low axial myopia and 20 donors with high axial myopia and found that the abundance of 16 metabolites, including many amino acids, nicotinoylglycine and hydroxyhippuric acid, was significantly different between the two groups. With the same criterion of AL, Barbas-Bernardos et al. [36] employed LC MS to screen the metabolite differences in the AHs of high and low myopia and identified 10 metabolites that were upregulated in the high-myopia group, such as L-arginine, citrulline, aminooctanoic acid and pantothenic acid, and 11 metabolites that were downregulated in this group, such as dihydropteoic acid, dimethylnonanoyl carnitine and aminocyclohexanecarboxylic. These studies working on myopia-related biomarkers in AH, however, were generally achieved by screening tests, and the indicative significance with myopia was not well verified by accurate quantification in a large cohort. Although there was less direct observation regarding SHs and myopia, some indirect evidence implicated the involvement of SHs in myopia. Certain glaucoma was induced by exogenous glucocorticoid administration or excess endogenous glucocorticoids from patients suffering from Cushing’s syndrome [19,37,38,39], while myopia was accompanied by an increased risk of primary open-angle glaucoma [40]. Herein, our study claimed that the abundance of the four SHs in AH, COR, CORT, A2 and ALD, in the myopia donors with AL ≥ 24mm was significantly higher than that in the donors with AL < 24 mm; moreover, the combination of these SHs could generate a discriminator to judge myopia. This means that SH abundance in the AH could serve as a molecular biomarker to predict AL shift or myopia incidence. Furthermore, the tight correlation of SHs in AH and AL could offer a partial explanation for why myopia occurs in glucocorticoid-induced glaucoma. Similar to the clinical significance of SHs in plasma, the ophthalmological application of SHs in AH is highly expected.

There are still some unsatisfied questions in this study. First of all, detection sensitivity by LC MS/MS is still an debatable for monitoring metabolites at low abundance. Although some MS/MS signals of SHs were detected, these signals were finally given up due to their poor MS/MS peaks and lower reproducibility. Second, our study indeed provides the first evidence of an SH profile in human AH and indicates the correlation between axial myopia and SHs. However, the observation has not yet been confirmed by other approaches. Even though the enzyme-linked immunosorbent assay (ELISA) is unable to global quantify SHs, the presence of SHs in AH should be easily detected by ELISA. Third, this study reveals some specific features of BAB based on the observation of the disproportionate distribution of SHs between plasma and AH. However, there is a lack of additional measurements to evaluate the SH transfer capacity through BAB in our study.

## 5. Conclusions

In summary, an LC–MS/MS-based approach was deployed to globally quantify SHs in human AH, and the first profile of the AH SHs was established, in which a total of six SHs were quantified and ranked in abundance order, F, COR, CORT, ALD, A2 and 1DOC. The AH and plasma SH profiles from 107 donors were quantified in parallel. The SH types and quantities in human plasma were overall higher than that in AH, while DHEAS, which is the most abundant plasma SH, was not measurable in AH. The observation led to the conclusion that BAB may play a regulatory role in transferring SHs and balancing the SH distribution between plasma and AH. Considering axial myopia resulted from all of the microenvironment changes in the eye, the AH metabolites are assumed to be indicators or regulators. The correlation of the AH SHs and axial myopia, hence, was systematically appraised, resulting in a panel consisting of five SHs as a discriminator for axial myopia and normal vision. Although the physiological significance of the AH SHs has not been explored yet, this study offers a clue that the AH SHs can serve as a molecular biomarker of axial myopia.

## Figures and Tables

**Figure 1 metabolites-12-01220-f001:**
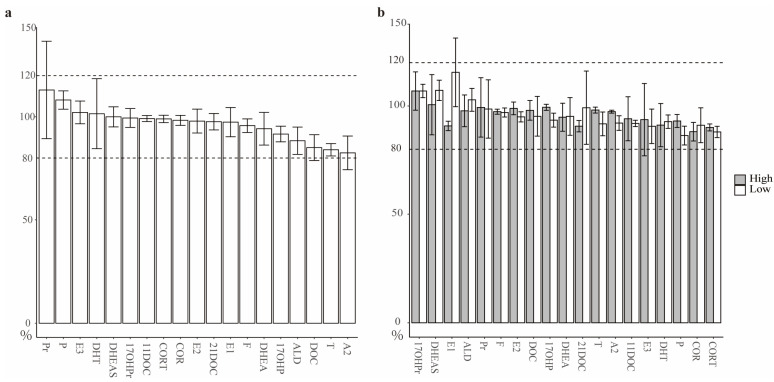
Assessment of data quality for quantification of the steroid hormones in AH using LC MS/MS. (**a**) The matrix effect of AH on steroid hormones. (**b**) The recoveries of steroid hormone extraction at high (solid bar) and low (hollow bar) concentration levels. The upper dashed line represents the recovery at 120%, and the lower dashed line represents the recovery at 80%.

**Figure 2 metabolites-12-01220-f002:**
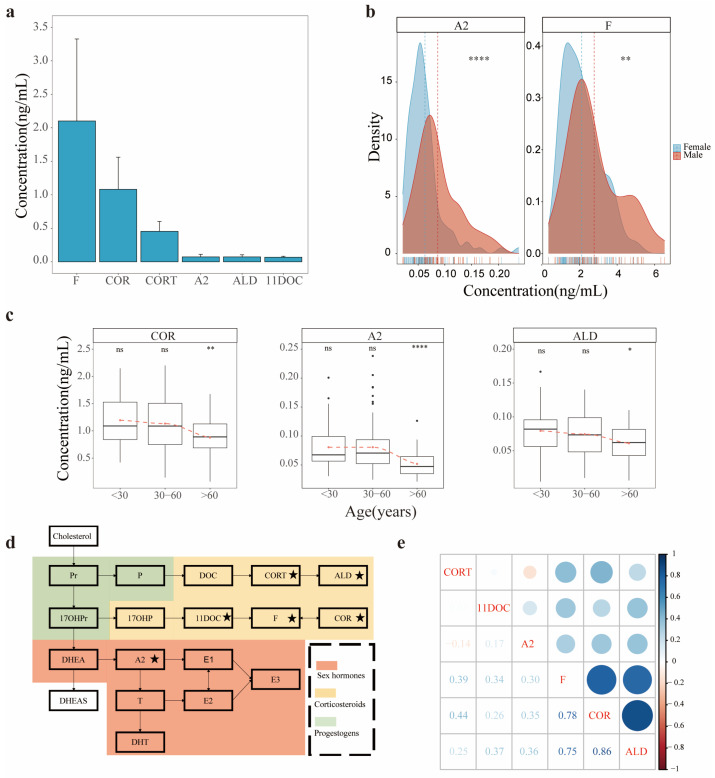
Profile of steroid hormones in AH. (**a**) The abundance distribution of steroid hormones in AH in this study. (**b**) Comparison of the A2 and F abundance distribution between male and female AH in the cohort this study. (** represents the *p* value < 0.01, and **** represents the *p* value < 0.0001). (**c**) The age-dependent potential of the 3 AH steroid hormones COR, A2 and ALD (* represents a *p* value < 0.05; ** represents a *p* value < 0.01, and **** represents a *p* value < 0.0001). (**d**) The metabolic pathways of steroid hormones in humans (the stars represent the steroid hormones detected in AH in this study). (**e**) The abundance correlations for all of the AH steroid hormones (Spearman’s correlation, the values represent Spearman’s correlation coefficients).

**Figure 3 metabolites-12-01220-f003:**
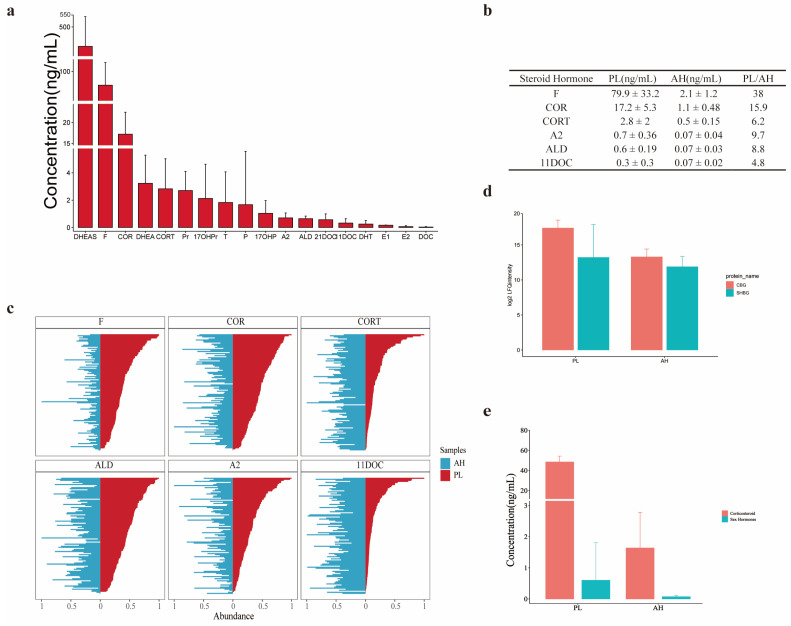
Comparison of steroid hormones in plasma and AH. (**a**) The abundance distribution of steroid hormones in plasmas in this study. (**b**) Table: The abundance of steroid hormones coidentified in plasma and AH in this study. (**c**) Individual comparison of the abundance ranking for the coidentified steroid hormones in plasmas and AHs in this cohort. (**d**) Comparison of the CBG and SHBG abundance between plasma and AH. (**e**) Comparison of the abundance sum of glucocorticoids or sexual steroid hormones between plasma and AH.

**Figure 4 metabolites-12-01220-f004:**
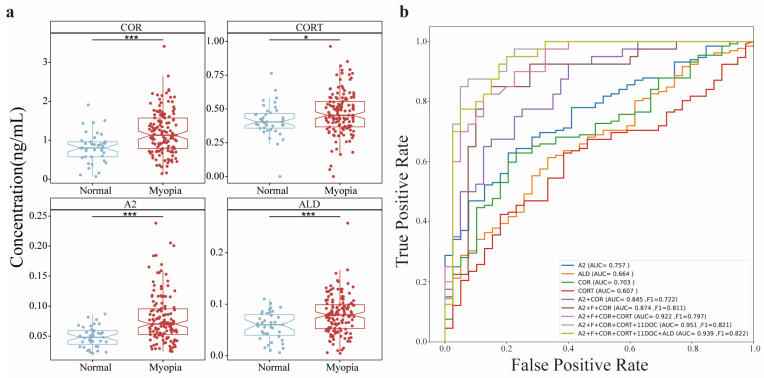
Correlation analysis of the axial length of the eye and the abundance of AH steroid hormones. (**a**) Comparison of the abundance of the 4 steroid hormones COR, CORT, A2 and ALD between the two groups with axial lengths ≥ 24 mm and <24 mm (* represents *p* value < 0.05, and *** represents *p* value < 0.001). (**b**) Discriminative prediction of the axial length of the eye based on the abundance of AH steroid hormones using the SVM model.

**Table 1 metabolites-12-01220-t001:** Clinical information of donors.

Case Number	Case Group	Age	Gender	Eye	Diopter	Degree of Turbidity	AL (od mm)	AL (os mm)	Correspond Plasma
20201225P0037	ARC	73	female	od	−3.25DC*177	C4N4P3	22	21.8	yes
20201225P0002	ARC	85	female	od	−1.50DC*94	C3N3P2	23.27	23.24	yes
20201225P0005	ARC	53	female	od	−3.75DS − 0.50DC*92	C2N3P4	22.24	22.11	yes
20201225P0067	ARC	79	male	od	−0.25DS − 0.70DC*5	C2N3P2	23.74	23.82	yes
20201225P0149	ARC	68	female	os	−0.25DS − 0.60DC*9	C2N3P2	23.07	23.05	yes
20201225P0041	ARC	69	female	od	−0.25DS − 0.30DC*92	C3N2P2	22.34	22.52	yes
20201225P0012	ARC	84	male	od	−0.75DC*82	C3N4P3	23.58	23.56	yes
20201225P0034	ARC	77	male	od	−2.00DS − 2.50DC*48	C2N3P2	24.2	23.84	yes
20201225P0003	ARC	82	male	od	−2.00DS − 3.00DC*110	C2N3P2	24.11	24.29	yes
20201225P0152	ARC	73	male	os	+2.00DS − 3.25DC*70	C4N3P3	24.16	23.95	yes
20201225P0137	ARC	91	male	os	−4.5DC*90	C4N4P4	22.54	22.56	yes
20201225P0121	ARC	65	female	os	+1.50DC*172	C3N2P2	23	23.12	yes
20200918P0020	ARC	67	female	od	−0.25DC*132	C2N2P2	24.62	24.28	yes
20200918P0013	ARC	71	female	od	+0.75DS + 1.00DC*90	C2N3P2	23.4	23.66	yes
20200918P0030	ARC	76	male	od	−1.50DS − 2.00DC*96	C3N3P3	23.12	23.41	yes
20200918P0019	ARC	73	male	od	+0.75DS + 1.00DC*180	C3N3P3	24.29	24.17	yes
20200918P0014	ARC	77	female	os	−0.50DC*15	C2N3P2	23.39	23.55	yes
20200918P0032	ARC	74	male	od	−1.50DS	C3N3P3	23.39	23.33	yes
20200918P0044	ARC	78	male	od	−0.50DC*135	C4N4P2	23.56	23.7	
20200918P0046	ARC	50	male	os	−0.75DC*77	C3N3P3	23.33	23.58	
20201225P0015	ARC	61	female	od	−-0.25DC*50	C2N3P2	22.01	22.1	
20201225P0147	ARC	80	female	os	−0.25DC*149	C3N3P2	24.6	24.72	
20201225P0072	ARC	69	female	os	+3.00DS − 3.00DC*25	C4N4P3	24.54	24.8	
20201225P0140	ARC	62	female	od	+2.00DS	C2N2P2	22.56	22.61	
20201225P0160	ARC	61	male	os	−1.25DC*77	C2N3P2	24.11	23.92	
20201225P0033	ARC	83	male	os	−2.00DC*85	C4N3P3	23.3	23.27	
20201225P0043	ARC	64	female	os	−0.75DC*153	C2N3P2	23.66	23.63	
20201225P0095	ARC	66	male	od	+1.25DS + 0.25DC*75	C3N3P2	23.99	23.88	
20201225P0009	ARC	63	female	od	+3.00DS	C2N3P2	23.74	23.32	
20201225P0071	ARC	80	male	od	−2.50DC*111	C2N3P2	22.22	22.8	yes
20201225P0019	ARC	55	female	od	−0.50DC*160	C4N4P3	24.47	24.34	yes
20201225P0133	ARC	79	female	od	−0.77DC*101	C2N3P2	22.43	22.33	yes
20201225P0065	ARC	59	male	od	−1.75DS − 0.75DC*85	C2N2P2	23.94	23.93	yes
20201225P0081	HM	27	male	od	−6.00DS − 2.50DC*5		27.99	24.48	yes
20201225P0045	HM	30	female	od	−8.37DS − 2.37DC*178		27.05	26.23	yes
20200918P0015	HM	30	female	od	−7.00DS − 1.25DC*172		25.09	25.31	yes
20200918P0003	HM	27	male	od	−5.75DS		26.66	26.68	yes
20200918P0037	HM	27	female	od	−5.00DS − 1.25DC*80		26.76	26.67	yes
20200918P0022	HM	36	female	od	−11.62DS − 1.12DC*16		27.2	27.34	yes
20200918P0017	HM	26	female	od	−6.12DS − 0.87DC*6		25.56	25.48	yes
20200918P0048	HM	21	female	od	−7.25DS − 0.87DC*179		25.47	25.48	
20200918P0043	HM	20	male	od	−14.00DS − 4.50DC*90		29.92	28.42	
20201225P0029	HM	36	female	od	−7.25DS		26.92	30.27	
20201225P0013	HM	30	female	od	−5.62DS − 0.37DC*51		26.16	26.01	
20201225P0135	HM	25	female	od	−5.87DS − 1.62DC*175		26.51	27.07	
20201225P0134	HM	23	female	od	−8.37DS − 1.00DC*4		27.86	27.77	
20201225P0079	HM	45	female	od	−6.25DS − 3.50DC*176		25.73	25.58	
20201225P0151	HM	36	male	od	−5.50DS		26.84	26.77	yes
20201225P0083	HM	19	male	od	−5.87DS − 0.37DC*46		26	26	yes
20201225P0111	HM	28	female	od	−10.50DS		27.96	27.19	yes
20201225P0044	HM	31	female	od	−9.00DS − 5.50DC*180		26.95	26.98	yes
20201225P0153	HM	32	female	od	−13.50DS		30.18	29.81	yes
20201225P0061	HM	24	female	od	−11.25DS − 1.50DC*10		26	25.82	yes
20201225P0105	HM	25	female	od	−5.25DS		26.32	26.24	yes
20201225P0008	HM	26	female	od	−6.75DS − 0.25DC*93		25.49	25.51	yes
20201225P0051	HM	39	female	od	−21.00DS − 3.00DC*10		25.79	26.35	yes
20201225P0084	HM	33	male	od	−7.00DS		25.79	26.35	yes
20201225P0040	HM	18	male	od	−8.62DS − 1.87DC*180		26.66	26.73	yes
20201225P0023	HM	35	female	od	−6		26	26	yes
20201225P0154	HM	28	male	od	−12.50DS − 0.75DC*228		28.95	28.49	yes
20201225P0128	HM	23	female	od	−10.50DS		26	26	yes
20201225P0062	HM	23	male	od	−12.50DS − 3.50DC*158		28.89	29.1	yes
20201225P0110	HM	23	female	od	−9.25DS − 0.62DC*2		27	27.5	yes
20201225P0001	HM	28	female	od	−6.75DS		25.36	25.22	yes
20201225P0122	HM	28	male	od	−10.00DS − 2.50DC*170		27.67	27.34	yes
20201225P0060	HM	19	male	od	−9.50DS − 1.75DC*180		26	26	yes
20201225P0119	HM	23	female	od	−10.00DS − 2.00DC*180		28.83	29.19	yes
20201225P0139	HM	26	male	od	−5.25DS − 2.25DC*174		27.13	26.98	yes
20201225P0118	HM	42	female	od	−7.62DS − 0.25DC*21		26.43	26.41	yes
20200918P0021	HM	24	female	od	−7.50DS − 0.50DC*2		26	26	yes
20200918P0012	HM	27	female	od	−8.12DS − 1.00DC*5		26.22	25.63	yes
20200918P0008	HM	24	female	od	−6.87DS − 2.87DC*6		25.97	26.13	yes
20200918P0039	HM	27	female	od	−5.12DS − 0.37DC*39		26.18	26.11	yes
20200918P0036	HMC	51	female	od	−8.00DS − 0.75DC*110	C2N3P2	26.79	26.68	yes
20200918P0004	HMC	52	male	od	−16.00DS − 0.75DC*4	C2N2P2	30.61	29.17	yes
20200918P0035	HMC	53	male	os	−8.50DS − 1.50DC*95	C2N2P2	30.32	28.9	yes
20200918P0026	HMC	53	male	od	−11.00DS − 2.00DC*70	C2N3P2	28.4	27.98	yes
20200918P0007	HMC	53	female	od	−7.50DS − 1.00DC*90	C2N2P2	26.8	26.22	yes
20200918P0018	HMC	31	male	os	−9.25DS	C2N2P1	25.95	25.89	yes
20200918P0011	HMC	51	male	os	−10.00DS − 0.50DC*25	C4N4P3	28.16	28.82	yes
20200918P0001	HMC	46	female	os	−9.00DS/ − 0.50DC*60	C2N2P2	26.63	25.77	yes
20200918P0005	HMC	50	female	od	−10.00DS	C2N2P2	28.21	28.67	yes
20200918P0028	HMC	42	male	os	−9.00DS − 0.750DC*25	C3N2P3	25.36	27.7	yes
20200918P0034	HMC	47	male	os	−11.25DS − 1.50DC*143	C2N2P2	28.9	28.15	yes
20200918P0038	HMC	44	female	od	−2.75DS	C2N2P2	25.89	25.7	yes
20200918P0002	HMC	54	male	os	−7.75DS − 2.25DC*174	C2N2P2	27.22	27.6	yes
20200918P0016	LM	22	female	od	−5.87DS − 0.50DC*2		25.89	25.83	yes
20200918P0047	LM	30	female	od	−5.50DS − 0.25DC*2		24.89	24.88	
20201225P0066	LM	26	female	od	−4.50DS − 1.00DC*10		24.78	25.02	
20201225P0014	LM	35	female	od	−4.75DS − 1.12DC*150		24.6	24.16	
20201225P0097	LM	26	female	od	−7.00DS		24.45	24.46	
20201225P0059	LM	34	male	od	−4.50DS − 0.37DC*136		25.32	25.37	
20201225P0021	LM	30	female	od	−5.00DS − 1.50DC*60		24.96	24.4	
20201225P0098	LM	27	female	od	−4.12DS − 0.87DC*13		24.91	24.38	
20201225P0126	LM	24	female	od	−5.50DS − 1.00DC*175		26.08	25.83	
20201225P0068	LM	26	female	od	−4.50DS		25.14	25.09	
20201225P0017	LM	31	female	od	−5.25DS		25.31	25.34	
20201225P0082	LM	37	female	od	−3.75DS		23.99	23.93	
20201225P0016	LM	18	female	od	−5.50DS − 1.50DC*2		25.72	26.12	
20201225P0091	LM	26	female	od	−3.50DS − 1.00DC*171		24.08	24.65	yes
20201225P0093	LM	29	female	od	−5.50DS − 0.62DC*87		25.27	25.77	yes
20201225P0094	LM	22	male	od	−7.75DS − 1.25DC*8		24.99	24.61	yes
20200918P0027	LM	33	female	od	−5.00DS		23	23	yes
20200918P0031	ARC	68	male	od	−0.25DC*45	C2N3P3	23.78	23.33	yes
20200918P0040	ARC	60	male	os	−0.50DC*50	C2N2P2	24.63	24.78	yes
20201225P0030	ARC	66	male	od	+0.90DC*104	C3N3P2	24.03	23.6	yes
20201225P0052	ARC	73	female	od	−0.50DC*86	C3N4P2	22.67	22.86	yes
20201225P0116	ARC	83	male	os	−1.75DC*91	C2N3P3	23.58	23.58	yes
20201225P0108	ARC	84	female	od	−1.00DC*133	C3N4P2	22.62	22.52	yes
20201225P0085	ARC	73	female	od	−1.00DC*78	C2N3P2	22.6	22.69	yes
20201225P0109	ARC	89	male	od	−1.50DC*78	C3N3P3	24.35	24.04	yes
20201225P0039	ARC	53	female	os	−0.25DS − 0.34DC*100	C1N3P3	23.9	23.66	yes
20201225P0022	ARC	75	female	os	+1.25DS + 1.50DC*170	C2N3P2	23.26	23.29	yes
20201225P0101	ARC	65	male	od	−0.25DS − 0.25DC*67	C2N4P2	23.46	23.2	yes
20200918P0051	ARC	68	female	od	−1.50DC*45	C4N4P4	23.71	23.49	
20201225P0115	ARC	91	female	os	−1.50DC*92	C4N4P4	23.44	22.95	
20201225P0114	ARC	82	female	od	−0.50DC*45	C4N3P3	23.4	23.38	
20201225P0027	HM	28	female	od	−6.37DS − 0.87DC*2		26.64	26.56	
20201225P0092	HM	27	male	od	−11.00DS − 3.00DC*5		29.61	28.36	
20201225P0130	HM	31	female	od	−8.37DS − 1.25DC*2		28.64	27.67	
20201225P0078	HM	21	male	od	−14.50DS − 0.25DC*7		29.32	28.78	
20201225P0018	HM	26	female	od	−10.00DS − 0.25DC*25		28.38	28.17	
20201225P0035	HM	23	male	od	−8.00DS − 3.00DC*175		29.12	28.76	yes
20200918P0029	HM	49	female	od	−10.25DS		26	26	yes
20200918P0041	HM	26	male	od	−7.50DS − 1.75DC*175		27.31	27.43	yes
20201225P0049	HM	20	female	od	−7.25DS − 1.25DC*38		27.92	25.17	yes
20201225P0020	HM	39	female	od	−4.75DS		26.71	26.74	yes
20201225P0090	HM	40	male	od	−4.75DS		26.49	26.67	yes
20201225P0142	HMC	58	male	od	−5.50DS − 0.50DC*90	C2N2P2	25.64	25.67	yes
20201225P0129	HMC	54	male	od	−4.25DS − 3.00DC*170	C2N3P3	26.53	26.15	yes
20200918P0045	HMC	52	male	os	−5.00DS	C1N1P4	26.74	26.51	
20200918P0050	HMC	72	female	od	−15.00DS − 0.50DC*85	C3N4P2	29.72	26.94	
20201225P0155	HMC	57	male	os	−7.00DS − 1.00DC*75	C2N2P2	27.14	26.92	
20201225P0103	HMC	37	male	od	−25.75DS	C2N2P3	33.8	25.74	
20201225P0031	HMC	59	female	os	−9.25DS − 0.50DC*16	C3N2P2	28	29.17	
20201225P0024	HMC	55	female	od	−9.00DS − 0.25DC*25	C3N4P2	28.05	25.34	
20201225P0074	HMC	52	male	od	−8.50DS − 0.75DC 167	C2N2P2	27.75	24.6	
20201225P0010	HMC	41	male	od	−8.00DS − 1.00DC*5	C2N2P2	26.05	26.12	
20201225P0146	HMC	58	female	os	−16.00DS − 1.50DC*121	C2N4P4	27.17	30.06	
20201225P0080	HMC	62	male	os	−7.50DS − 1.00DC*75	C2N2P2	25.79	26.32	
20201225P0026	HMC	68	female	os	−7.00DS − 1.50DC*175	C3N2P2	27.06	25.73	
20201225P0050	HMC	74	male	os	−20.00DS − 1.50DC*155	C2N2P2	31.31	31.31	
20201225P0028	HMC	48	male	os	−14.25DS − 2.25DC*80	C2N3P2	27.42	27.48	
20201225P0025	HMC	76	female	os	−7.50DS − 0.75DC*80	C2N2P2	27.6	28.15	
20201225P0100	HMC	54	female	os	−23.00DS	C2N2P2	32.49	31.5	yes
20201225P0124	HMC	75	male	os	−12.00DS − 2.50DC*65	C3N3P2	28.98	28.99	yes
20201225P0104	LM	31	female	od	−2.75DS		24.57	24.47	yes
20201225P0148	LM	27	female	od	−5.00DS		25.5	25.37	yes
20201225P0141	LM	28	female	od	−5.25DS − 2.00DC*5		25.07	25.96	yes
20201225P0058	LM	24	male	od	−3.25DS		25.36	25.39	yes
20201225P0159	LM	36	female	od	−5.00DS		25.61	25.47	yes
20201225P0088	LM	30	female	od	−4.75DS		24.81	24.85	yes
20201225P0145	LM	23	female	os	−3.75DS		25.86	25.86	yes
20201225P0158	LM	24	female	od	−5.25DS		24.41	24.51	yes
20201225P0138	LM	24	female	od	−5.00DS		23	23	yes
20200918P0025	LM	31	female	od	−5.25DS − 1.00DC*12		24.68	24.12	yes
20200918P0006	HM	20	male	od	−10.87DS − 4.37DC*1		30.9	27.88	yes
20200918P0024	HMC	37	female	od	−23.00DS	C1N1P3	32.04	30.84	yes
20200918P0023	HMC	41	female	od	−25.00DS − 1.50DC*162	C3N4P2	33.64	32.41	yes
20200918P0033	HMC	60	male	od	−21.50DS	C3N3P2	28.68	29.31	
20200918P0042	HMC	49	female	od	−24.00DS − 2.25DC*135	C2N2P2	32.6	31.87	
20201225P0069	HMC	51	male	od	−29.00DS	C2N2P2	31.48	30.97	
20201225P0156	HMC	53	female	os	−20.00DS − 1.00DC*70	C2N2P2	31.49	31.17	
20201225P0064	HMC	46	male	od	−14.00DS	C2N2P2	28.8	28.56	
20201225P0113	HMC	58	female	od	−6.00DS	C3N2P3	26.34	25.74	
20201225P0125	HMC	69	male	os	−8.5DS − 1.50DC*85	C2N2P2	31.42	30.18	
20201225P0099	HMC	54	male	od	−15.50DS − 1.50DC*20	C2N3P2	25.93	25.96	
20201225P0004	HMC	52	female	os	−15.00DS − 0.50DC*7	C2N2P2	29.07	30.06	
20201225P0144	HMC	50	female	os	−3.00DS	C2N3P2	26.83	26.73	
20201225P0057	HMC	55	male	od	−11.00DS	C2N2P2	31.18	32.42	
20201225P0165	HMC	63	male	os	−22.50DS	C5N4P2	28.94	28.85	

## Data Availability

All the origin spectrum data of AHs in the research is available through the China National GeneBank Sequence Archive (CNSA) (https://db.cngb.org/cnsa/) (accessed on 21 September 2022) with accession number CNP0003508.

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
