# Peer review of "First Evidence Indicates the Physiology- and Axial-Myopia-Dependent Profiles of Steroid Hormones in Aqueous Humor"

_metabolites, 2022, doi:10.3390/metabo12121220_

Round 1
Reviewer 1 Report
In the present work, the authors first detected the concentrations of steroids hormones (SHs) in the plasma and aqueous humor of patients with cataract or axial myopia. This is a quite interesting and important work for us to understand the physiological roles of SHs in aqueous humor and eye diseases. The study was well designed and analyzed. However, there are several improvements should be made prior to publication
1. Abstract: It is necessary to explain the correlation between SHs in aqueous humor and serum and ocular axis length. Otherwise, the conclusion of ‘……can potentially contribute to axial myopia’ appears abrupt.
2. In Section 2.2, you should offer the clinical study design scheme and the clinical trial register number. What is the inclusion and exclusion criteria?
3. In Section 2.2, how you identify normal axial length and long axial length?
4. In Section 2.2, It may be better to categorize senile cataract into three subgroups: cortical cataract, age-related nuclear cataract, and posterior subcapsular cataract. The result should also be re-calculated.
5. In Section 2.5, does aqueous humor dilution affect the accuracy of LC MS/MS measurement?
6. In Section 2.8, what are the data present? Mean ± SD or SEM?
7. Figure 3, the picture definition needs to be improved. It is recommended to change it to vector image.
8. In Section 2.8, line 382 to 386, should be replaced to Method section.
9. In Discussion section, the limitation of the study should be added in the last paragraph.
10. Conclusion should be separated into segments, and indicate the guiding significance of this study for clinical practice.

Author Response
I appreciate you very much to deliver constructive criticism to our manuscript (metabolites-2002783), which is critically important to improving the manuscript's quality. I have carefully read all the comments and made the corresponding changes.
Please see the attachment.

Reviewer 2 Report
Lines 2-3: Please indicate the study’s design with a commonly used term in the title and/or in abstract
Introduction:
Line 40: Please indicate the reference of Deborah and Stefan right after these names
lines 52-53: Please indicate the reference of Rosa et al. right after these names
line 55: Please indicate the reference of Tina et al. right after these names and do it this way in the rest of the manuscript
An the end of the introdutction, please state any prespecified hypotheses related with the study objectives
Methods:
lines 100-109: Please describe the eligibility criteria, and the sources and methods of selection of participants and indicate the periods of recruitment
lines 182-190: Did the authors perform any test to check whether the variables follow a normal distribution? Please state it before indicating that non-parametric tests were used
Did the authors determine the minimum sample size required for the study?
Results:
Please provide a Table with the characteristics of study participants (eg demographic, clinical, refractive status, axial length)
Please indicate number of participants with missing data for each variable of interest
lines 193-195: In my opinion, this paragraph should be included in the methods section
lines 249-250: In my opinion, this criterion should be described in the methods section and needs a reference or a justification
ines 267-270: In my opinion, the criterion for dividing the sample by age should be described in the methods section and needs a reference or a justification
lines 382-384: In my opinion, the criterion for dividing the sample by axial lenght should be described in the methods section and needs a reference or a justification. Furthermore, besides the axial lenghtn did your have information about the refractive status?
Discussion:
Discuss limitations of the study, taking into account sources of potential bias or imprecision
Discuss the generalisability (external validity) of the study results
Please add a conclusion paragraph at the end of the study
Author Response

(The authors gave the same response as above.)

Reviewer 3 Report
The present research finding reported in the manuscript is an important addition to the filled of ocular research as it is specifically reporting the presence of signature steroid hormones (SHs) SHs (F, COR, CORT, ALD and A2) in the aqueous humor (AH) that are previously been linked to axial myopia. The plasma concentration of those SHs are higher than that found in the AH. Interestingly, no dehydroepiandosterone found in AH indicating the protection provided by the blood aqueous barrier (BAB). The authors concluded that the differential distribution of SHs in the AH could be indicative of axial myopia.
I have two a couple concerns:
1. Incorporating the subtype of myopia (Axial myopia) in the abstract section and throughout the manuscript is confusing. The authors need to introduce the readers to the basic classification of myopia and what the types of myopia are. The title of the manuscript does not talk about axial myopia. It just mentions myopia.
2. Where is the ethical approval (IRB) of the protocols that have been used to collect AH samples from the patients? Do they provide written informed consent?
3. Show a graphical presentation of the sex dependent distribution of the different SHs found in both AH and in plasma.
Minor:
1. Cite references based on the context and the publication. Do not cite all the references together in one place after mentioning different findings together. Example Page1, lines 40-43: ‘Deborah and Stefan claimed that ……while Ye et al. constructed a discriminator for 3 CAH subtypes using SHs in blood [10, 11].’ Cite Deborah’s paper weher her name is mentioned and cite Ye’s paper where his name is mentioned. Follow my suggestions for all the references cited in the manuscript.
2. Made the manuscript more concised and focused. In its present form the manuscript is unusually long.
3. Italicize et al used throughout.
Author Response

(The authors gave the same response as above.)

Round 2
Reviewer 1 Report
The author has revised the manuscript as required, and I recommended to accept the manuscript
Author Response
We're really grateful and appreciate you taking the time to share your advice with us. And we are grateful for your approval of the revised manuscript. Thank you
Reviewer 2 Report
Title:
I agree that the title does not contain unusual terms. A point I made in my previous review was that the study design (i.e. pilot study, observational study, etc.) should be mentioned in the title and/or abstract.
Introduction:
To improve readability, the reference numbers should be placed right after the author's name, not at the end of the citation. For example: Tina et al. (19)
Methods:
Please clearly state the definition for each group: High myopia vs. Low miopia. Also, the difference in axial leght does not imply a difference in refractive status (subjects with axial length over 24 might not be myopes) so this groups indicate difference in axial length, not in refractive status
The methods of recrutiment are not described, just the sources
In the author's reply, they indicate "we did the Shapiro-Wilk’s test before Hypothesis test to look at whether the dataset fit the normal distribution or not. And if the p value of Shapiro-Wilk’s test is less than 0.05, the dataset is not in normal distribution. In this study, most of p values for 6 SHs dataset in different comparison were less than 0.05, so we chose Hypothesis test with non-parametric setting" but it is not mentioned in the manuscript
The assumptions and calculations done for clculating the minimum sample size required for the study should be clearly stated in the manuscript
Results:
To improve readability, the table with the characteristics of study participants should be included in the text, not as supplementary material
Please indicate in the text that there were no missing data
The association between axial length and refractive status is clear but the authors divide the sample according to the axial length, not the refractive status, so they should describe differences according to the axial length, not refractive status.
Author Response
I appreciate you very much for your constructive criticism of our manuscript. I have carefully read all the comments while following your suggestions, and I have made the corresponding changes, point by point, which are attached to this letter.

Reviewer 3 Report
In the revised manuscript the authors have mostly addressed my comments and concerns except for the statement of ethical use of human AH samples.
Page 3, line 129: ‘all the donors have received the written informed consent.’ This statement needs to be rephrased and corrected. Donors have received the written consent form, but it is not clear whether they have signed the form. A better way of phrasing the statement could be ‘The donors were informed about the sample collection and they have signed the informed consent forms.’
Author Response
In this two-round review, I greatly appreciate your constructive criticism of our manuscript. I have made the corresponding changes to your suggestion.
- Page 3, line 129: ‘all the donors have received the written informed consent.’ This statement needs to be rephrased and corrected. Donors have received the written consent form, but it is not clear whether they have signed the form. A better way of phrasing the statement could be ‘The donors were informed about the sample collection and they have signed the informed consent forms.’
Thanks a lot for your suggestion. I have changed the description.